# Deep Gradient Compression: Reducing the Communication Bandwidth for Distributed Training

**Yujun Lin** [*]
Tsinghua University
Beijing National Research Center
for Information Science and Technology
`linyy14@mails.tsinghua.edu.cn`

**Song Han** [†]
Stanford University
Google Brain
`songhan@stanford.edu`

**Huizi Mao**
Stanford University
`huizi@stanford.edu`

**Yu Wang**
Tsinghua University
Beijing National Research Center
for Information Science and Technology
`yu-wang@mail.tsinghua.edu.cn`

**William J. Dally**
Stanford University
NVIDIA
`dally@stanford.edu`

## Abstract

Large-scale distributed training requires significant communication bandwidth for gradient exchange that limits the scalability of multi-node training, and requires expensive high-bandwidth network infrastructure. The situation gets even worse with distributed training on mobile devices (federated learning), which suffers from higher latency, lower throughput, and intermittent poor connections. In this paper, we find 99.9% of the gradient exchange in distributed SGD are redundant, and propose Deep Gradient Compression (DGC) to greatly reduce the communication bandwidth. To preserve accuracy during this compression, DGC employs four methods: momentum correction, local gradient clipping, momentum factor masking, and warm-up training. We have applied Deep Gradient Compression to image classification, speech recognition, and language modeling with multiple datasets including Cifar10, ImageNet, Penn Treebank, and Librispeech Corpus. On these scenarios, Deep Gradient Compression achieves a gradient compression ratio from $270\times$ to $600\times$ without losing accuracy, cutting the gradient size of ResNet-50 from 97MB to 0.35MB, and for DeepSpeech from 488MB to 0.74MB. Deep gradient compression enables large-scale distributed training on inexpensive commodity 1Gbps Ethernet and facilitates distributed training on mobile.

## 1 Introduction

Large-scale distributed training improves the productivity of training deeper and larger models (Chilimbi et al., 2014; Xing et al., 2015; Moritz et al., 2015; Zinkevich et al., 2010). Synchronous stochastic gradient descent (SGD) is widely used for distributed training. By increasing the number of training nodes and taking advantage of data parallelism, the total computation time of the forward-backward passes on the same size training data can be dramatically reduced. However, gradient exchange is costly and dwarfs the savings of computation time (Li et al., 2014; Wen et al.,

---

[*]Work done while at Stanford CVA lab.
[†]Joining MIT in 2018.

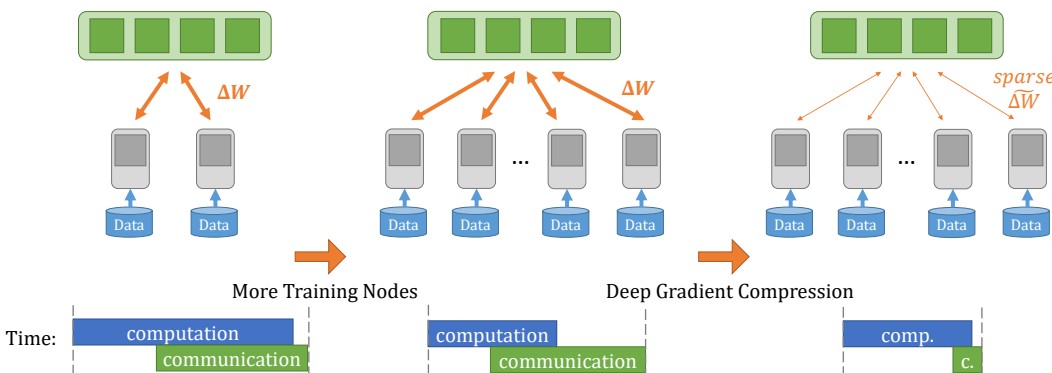

Figure 1: Deep Gradient Compression can reduce the communication time, improve the scalability, and speed up distributed training.

2017), especially for recurrent neural networks (RNN) where the computation-to-communication ratio is low. Therefore, the network bandwidth becomes a significant bottleneck for scaling up distributed training. This bandwidth problem gets even worse when distributed training is performed on mobile devices, such as federated learning (McMahan et al., 2016; Konečný et al., 2016). Training on mobile devices is appealing due to better privacy and better personalization (Google, 2017), but a critical problem is that those mobile devices suffer from even lower network bandwidth, intermittent network connections, and expensive mobile data plan.

Deep Gradient Compression (DGC) solves the communication bandwidth problem by compressing the gradients, as shown in Figure 1. To ensure no loss of accuracy, DGC employs *momentum correction* and *local gradient clipping* on top of the gradient sparsification to maintain model performance. DGC also uses *momentum factor masking* and *warmup training* to overcome the staleness problem caused by reduced communication.

We empirically verified Deep Gradient Compression on a wide range of tasks, models, and datasets: CNN for image classification (with Cifar10 and ImageNet), RNN for language modeling (with Penn Treebank) and speech recognition (with Librispeech Corpus). These experiments demonstrate that gradients can be compressed up to $600\times$ without loss of accuracy, which is an order of magnitude higher than previous work (Aji & Heafield, 2017).

## 2 RELATED WORK

Researchers have proposed many approaches to overcome the communication bottleneck in distributed training. For instance, asynchronous SGD accelerates the training by removing gradient synchronization and updating parameters immediately once a node has completed back-propagation (Dean et al., 2012; Recht et al., 2011; Li et al., 2014). Gradient quantization and sparsification to reduce communication data size are also extensively studied.

**Gradient Quantization** Quantizing the gradients to low-precision values can reduce the communication bandwidth. Seide et al. (2014) proposed 1-bit SGD to reduce gradients transfer data size and achieved $10\times$ speedup in traditional speech applications. Alistarh et al. (2016) proposed another approach called QSGD which balance the trade-off between accuracy and gradient precision. Similar to QSGD, Wen et al. (2017) developed TernGrad which uses 3-level gradients. Both of these works demonstrate the convergence of quantized training, although TernGrad only examined CNNs and QSGD only examined the training loss of RNNs. There are also attempts to quantize the entire model, including gradients. DoReFa-Net (Zhou et al., 2016) uses 1-bit weights with 2-bit gradients.

**Gradient Sparsification** Strom (2015) proposed threshold quantization to only send gradients larger than a predefined constant threshold. However, the threshold is hard to choose in practice. Therefore, Dryden et al. (2016) chose a fixed proportion of positive and negative gradient updates separately, and Aji & Heafield (2017) proposed Gradient Dropping to sparsify the gradients by a single threshold based on the absolute value. To keep the convergence speed, Gradient Dropping

**Algorithm 1** Gradient Sparsification on node $k$

---

**Input:** dataset $\chi$
**Input:** minibatch size $b$ per node
**Input:** the number of nodes $N$
**Input:** optimization function $SGD$
**Input:** init parameters $w = \{w[0], w[1], \cdots, w[M]\}$
1:  $G^k \leftarrow 0$
2:  **for** $t = 0, 1, \cdots$ **do**
3:      $G_t^k \leftarrow G_{t-1}^k$
4:      **for** $i = 1, \cdots, b$ **do**
5:          Sample data $x$ from $\chi$
6:          $G_t^k \leftarrow G_t^k + \frac{1}{Nb} \nabla f(x; w_t)$
7:      **end for**
8:      **for** $j = 0, \cdots, M$ **do**
9:          Select threshold: $thr \leftarrow s\%$ of $\left| G_t^k[j] \right|$
10:         $Mask \leftarrow \left| G_t^k[j] \right| > thr$
11:         $\widetilde{G}_t^k[j] \leftarrow G_t^k[j] \odot Mask$
12:         $G_t^k[j] \leftarrow G_t^k[j] \odot \neg Mask$
13:     **end for**
14:     $All\text{-}reduce \ G_t^k : G_t \leftarrow \sum_{k=1}^N encode(\widetilde{G}_t^k)$
15:     $w_{t+1} \leftarrow SGD\left(w_t, G_t\right)$
16: **end for**

---

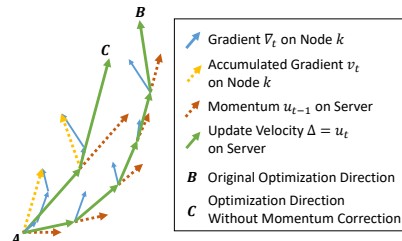

(a) Local Gradient Accumulation without momentum correction

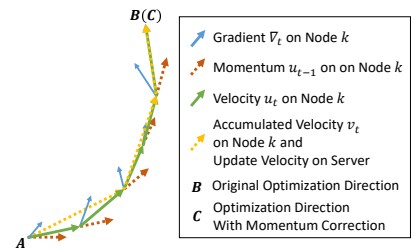

(b) Local Gradient Accumulation with momentum correction

Figure 2: Momentum Correction

requires adding the layer normalization(Lei Ba et al., 2016). Gradient Dropping saves 99% of gradient exchange while incurring 0.3% loss of BLEU score on a machine translation task. Concurrently, Chen et al. (2017) proposed to automatically tunes the compression rate depending on local gradient activity, and gained compression ratio around $200\times$ for fully-connected layers and $40\times$ for convolutional layers with negligible degradation of top-1 accuracy on ImageNet dataset.

Compared to the previous work, DGC pushes the gradient compression ratio to up to $600\times$ for the whole model (same compression ratio for all layers). DGC does not require extra layer normalization, and thus does not need to change the model structure. Most importantly, Deep Gradient Compression results in no loss of accuracy.

## 3  DEEP GRADIENT COMPRESSION

### 3.1  GRADIENT SPARSIFICATION

We reduce the communication bandwidth by sending only the important gradients (sparse update). We use the gradient magnitude as a simple heuristics for importance: only gradients larger than a threshold are transmitted. To avoid losing information, we accumulate the rest of the gradients locally. Eventually, these gradients become large enough to be transmitted. Thus, we send the large gradients immediately but eventually send all of the gradients over time, as shown in Algorithm 1. The $encode()$ function packs the 32-bit nonzero gradient values and 16-bit run lengths of zeros.

The insight is that the local gradient accumulation is equivalent to increasing the batch size over time. Let $F(w)$ be the loss function which we want to optimize. Synchronous Distributed SGD performs the following update with $N$ training nodes in total:

$$F(w) = \frac{1}{|\chi|} \sum_{x \in \chi} f(x, w), \qquad w_{t+1} = w_t - \eta \frac{1}{Nb} \sum_{k=1}^N \sum_{x \in \mathcal{B}_{k,t}} \nabla f(x, w_t) \tag{1}$$

where $\chi$ is the training dataset, $w$ are the weights of a network, $f(x, w)$ is the loss computed from samples $x \in \chi$, $\eta$ is the learning rate, $N$ is the number of training nodes, and $\mathcal{B}_{k,t}$ for $1 \le k < N$ is a sequence of $N$ minibatches sampled from $\chi$ at iteration $t$, each of size $b$.

Consider the weight value $w^{(i)}$ of $i$-th position in flattened weights $w$. After $T$ iterations, we have

$$w_{t+T}^{(i)} = w_t^{(i)} - \eta T \cdot \frac{1}{NbT} \sum_{k=1}^{N} \left( \sum_{\tau=0}^{T-1} \sum_{x \in \mathcal{B}_{k,t+\tau}} \nabla^{(i)} f(x, w_{t+\tau}) \right) \tag{2}$$

Equation 2 shows that local gradient accumulation can be considered as increasing the batch size from $Nb$ to $NbT$ (the second summation over $\tau$), where $T$ is the length of the *sparse update interval* between two iterations at which the gradient of $w^{(i)}$ is sent. Learning rate scaling (Goyal et al., 2017) is a commonly used technique to deal with large minibatch. It is automatically satisfied in Equation 2 where the $T$ in the learning rate $\eta T$ and batch size $NbT$ are canceled out.

## 3.2 IMPROVING THE LOCAL GRADIENT ACCUMULATION

Without care, the sparse update will greatly harm convergence when sparsity is extremely high (Chen et al., 2017). For example, Algorithm 1 incurred more than 1.0% loss of accuracy on the Cifar10 dataset, as shown in Figure 3(a). We find momentum correction and local gradient clipping can mitigate this problem.

**Momentum Correction**   Momentum SGD is widely used in place of vanilla SGD. However, Algorithm 1 doesn't directly apply to SGD with the momentum term, since it ignores the discounting factor between the sparse update intervals.

Distributed training with vanilla momentum SGD on $N$ training nodes follows (Qian, 1999),

$$u_t = m u_{t-1} + \sum_{k=1}^{N} (\nabla_{k,t}), \quad w_{t+1} = w_t - \eta u_t \tag{3}$$

where $m$ is the momentum, $N$ is the number of training nodes, and $\nabla_{k,t} = \frac{1}{Nb} \sum_{x \in \mathcal{B}_{k,t}} \nabla f(x, w_t)$.

Consider the weight value $w^{(i)}$ of $i$-th position in flattened weights $w$. After $T$ iterations, the change in weight value $w^{(i)}$ shows as follows,

$$w_{t+T}^{(i)} = w_t^{(i)} - \eta \left[ \cdots + \left( \sum_{\tau=0}^{T-2} m^{\tau} \right) \nabla_{k,t+1}^{(i)} + \left( \sum_{\tau=0}^{T-1} m^{\tau} \right) \nabla_{k,t}^{(i)} \right] \tag{4}$$

If SGD with the momentum is directly applied to the sparse gradient scenario (line 15 in Algorithm 1), the update rule is no longer equivalent to Equation 3, which becomes:

$$v_{k,t} = v_{k,t-1} + \nabla_{k,t}, \quad u_t = m u_{t-1} + \sum_{k=1}^{N} sparse(v_{k,t}), \quad w_{t+1} = w_t - \eta u_t \tag{5}$$

where the first term is the local gradient accumulation on the training node $k$. Once the accumulation result $v_{k,t}$ is larger than a threshold, it will pass hard thresholding in the $sparse()$ function, and be encoded and get sent over the network in the second term. Similarly to the line 12 in Algorithm 1, the accumulation result $v_{k,t}$ gets cleared by the mask in the $sparse()$ function.

The change in weight value $w^{(i)}$ after the sparse update interval $T$ becomes,

$$w_{t+T}^{(i)} = w_t^{(i)} - \eta \left( \cdots + \nabla_{k,t+1}^{(i)} + \nabla_{k,t}^{(i)} \right) \tag{6}$$

The disappearance of the accumulated discounting factor $\sum_{\tau=0}^{T-1} m^{\tau}$ in Equation 6 compared to Equation 4 leads to the loss of convergence performance. It is illustrated in Figure 2(a), where Equation 4 drives the optimization from point $A$ to point $B$, but with local gradient accumulation, Equation 4 goes to point $C$. When the gradient sparsity is high, the update interval $T$ dramatically increases, and thus the significant side effect will harm the model performance. To avoid this error, we need momentum correction on top of Equation 5 to make sure the sparse update is equivalent to the dense update as in Equation 3.

If we regard the velocity $u_t$ in Equation 3 as "gradient", the second term of Equation 3 can be considered as the vanilla SGD for the "gradient" $u_t$. The local gradient accumulation is proved to be effective for the vanilla SGD in Section 3.1. Therefore, we can locally accumulate the velocity $u_t$ instead of the real gradient $\nabla_{k,t}$ to migrate Equation 5 to approach Equation 3:

$$u_{k,t} = mu_{k,t-1} + \nabla_{k,t}, \quad v_{k,t} = v_{k,t-1} + u_{k,t}, \quad w_{t+1} = w_t - \eta \sum_{k=1}^{N} sparse\left(v_{k,t}\right) \quad (7)$$

where the first two terms are the corrected local gradient accumulation, and the accumulation result $v_{k,t}$ is used for the subsequent sparsification and communication. By this simple change in the local accumulation, we can deduce the accumulated discounting factor $\sum_{\tau=0}^{T-1} m^\tau$ in Equation 4 from Equation 7, as shown in Figure 2(b).

We refer to this migration as the *momentum correction*. It is a tweak to the update equation, it doesn't incur any hyper parameter. Beyond the vanilla momentum SGD, we also look into Nesterov momentum SGD in Appendix B, which is similar to momentum SGD.

**Local Gradient Clipping**  Gradient clipping is widely adopted to avoid the exploding gradient problem (Bengio et al., 1994). The method proposed by Pascanu et al. (2013) rescales the gradients whenever the sum of their L2-norms exceeds a threshold. This step is conventionally executed *after* gradient aggregation from all nodes. Because we accumulate gradients over iterations on each node independently, we perform the gradient clipping locally *before* adding the current gradient $G_t$ to previous accumulation ($G_{t-1}$ in Algorithm 1). As explained in Appendix C, we scale the threshold by $N^{-1/2}$, the current node's fraction of the global threshold if all $N$ nodes had identical gradient distributions. In practice, we find that the local gradient clipping behaves very similarly to the vanilla gradient clipping in training, which suggests that our assumption might be valid in real-world data.

As we will see in Section 4, momentum correction and local gradient clipping help improve the word error rate from 14.1% to 12.9% on the AN4 corpus, while training curves follow the momentum SGD more closely.

### 3.3   OVERCOMING THE STALENESS EFFECT

Because we delay the update of small gradients, when these updates do occur, they are outdated or *stale*. In our experiments, most of the parameters are updated every 600 to 1000 iterations when gradient sparsity is 99.9%, which is quite long compared to the number of iterations per epoch. Staleness can slow down convergence and degrade model performance. We mitigate staleness with momentum factor masking and warm-up training.

**Momentum Factor Masking**  Mitliagkas et al. (2016) discussed the staleness caused by asynchrony and attributed it to a term described as *implicit momentum*. Inspired by their work, we introduce *momentum factor masking*, to alleviate staleness. Instead of searching for a new momentum coefficient as suggested in Mitliagkas et al. (2016), we simply apply the same mask to both the accumulated gradients $v_{k,t}$ and the momentum factor $u_{k,t}$ in Equation 7:

$$Mask \leftarrow |v_{k,t}| > thr, \quad v_{k,t} \leftarrow v_{k,t} \odot \neg Mask, \quad u_{k,t} \leftarrow u_{k,t} \odot \neg Mask$$

This mask stops the momentum for delayed gradients, preventing the stale momentum from carrying the weights in the wrong direction.

**Warm-up Training**  In the early stages of training, the network is changing rapidly, and the gradients are more diverse and aggressive. Sparsifying gradients limits the range of variation of the model, and thus prolongs the period when the network changes dramatically. Meanwhile, the remaining aggressive gradients from the early stage are accumulated before being chosen for the next update, and therefore they may outweigh the latest gradients and misguide the optimization direction. The *warm-up training* method introduced in large minibatch training (Goyal et al., 2017) is helpful. During the warm-up period, we use a less aggressive learning rate to slow down the changing speed of the neural network at the start of training, and also less aggressive gradient sparsity, to reduce the number of extreme gradients being delayed. Instead of linearly ramping up the learning

Table 1: Techniques in Deep Gradient Compression

| Techniques | Gradient Dropping (Aji & Heafield, 2017) | Deep Gradient Compression | Reduce Bandwidth | Ensure Convergence | Overcome Staleness | |
| --- | --- | --- | --- | --- | --- | --- |
| | | | | | Improve Accuracy | Maintain Convergence Iterations |
| Gradient Sparsification | ✓ | ✓ | ✓ | - | - | - |
| Local Gradient Accumulation | ✓ | ✓ | - | ✓ | - | - |
| Momentum Correction | - | ✓ | - | - | ✓ | - |
| Local Gradient Clipping | - | ✓ | - | ✓ | - | ✓ |
| Momentum Factor Masking | - | ✓ | - | - | ✓ | ✓ |
| Warm-up Training | - | ✓ | - | - | ✓ | ✓ |

rate during the first several epochs, we exponentially increase the gradient sparsity from a relatively small value to the final value, in order to help the training adapt to the gradients of larger sparsity.

As shown in Table 1, momentum correction and local gradient clipping improve the local gradient accumulation, while the momentum factor masking and warm-up training alleviate the staleness effect. On top of gradient sparsification and local gradient accumulation, these four techniques make up the Deep Gradient Compression (pseudo code in Appendix D), and help push the gradient compression ratio higher while maintaining the accuracy.

## 4 EXPERIMENTS

### 4.1 EXPERIMENT SETTINGS

We validate our approach on three types of machine learning tasks: image classification on Cifar10 and ImageNet, language modeling on Penn Treebank dataset, and speech recognition on AN4 and Librispeech corpus. The only hyper-parameter introduced by Deep Gradient Compression is the warm-up training strategy. In all experiments related to DGC, we rise the sparsity in the warm-up period as follows: 75%, 93.75%, 98.4375%, 99.6%, 99.9% (exponentially increase till 99.9%). We evaluate the reduction in the network bandwidth by the gradient compression ratio as follows,

$$\text{Gradient Compression Ratio} = size\left[encode\left(sparse(G^k)\right)\right]/size\left[G^k\right]$$

where $G^k$ is the gradients computed on the training node $k$.

**Image Classification** We studied ResNet-110 on Cifar10, AlexNet and ResNet-50 on ImageNet. Cifar10 consists of 50,000 training images and 10,000 validation images in 10 classes (Krizhevsky & Hinton, 2009), while ImageNet contains over 1 million training images and 50,000 validation images in 1000 classes (Deng et al., 2009). We train the models with *momentum SGD* following the training schedule in Gross & Wilber (2016). The warm-up period for DGC is 4 epochs out of 164 epochs for Cifar10 and 4 epochs out of 90 epochs for ImageNet Dataset.

**Language Modeling** The Penn Treebank corpus (PTB) dataset consists of 923,000 training, 73,000 validation and 82,000 test words (Marcus et al., 1993). The vocabulary we select is the same as the one in Mikolov et al. (2010). We adopt the 2-layer LSTM language model architecture with 1500 hidden units per layer (Press & Wolf, 2016), tying the weights of encoder and decoder as suggested in Inan et al. (2016) and using *vanilla SGD* with gradient clipping, while learning rate decays when no improvement has been made in validation loss. The warm-up period is 1 epoch out of 40 epochs.

Table 2: ResNet-110 trained on Cifar10 Dataset

| # GPUs in total | Batchsize in total per iteration | Training Method | Top 1 Accuracy | |
|---|---|---|---|---|
| 4 | 128 | Baseline | 93.75% | |
| | | Gradient Dropping (Aji & Heafield, 2017) | 92.75% | -1.00% |
| | | Deep Gradient Compression | **93.87%** | **+0.12%** |
| 8 | 256 | Baseline | 92.92% | |
| | | Gradient Dropping (Aji & Heafield, 2017) | 93.02% | +0.10% |
| | | Deep Gradient Compression | **93.28%** | **+0.37%** |
| 16 | 512 | Baseline | 93.14% | |
| | | Gradient Dropping (Aji & Heafield, 2017) | 92.93% | -0.21% |
| | | Deep Gradient Compression | **93.20%** | **+0.06%** |
| 32 | 1024 | Baseline | 93.10% | |
| | | Gradient Dropping (Aji & Heafield, 2017) | 92.10% | -1.00% |
| | | Deep Gradient Compression | **93.18%** | **+0.08%** |

Table 3: Comparison of gradient compression ratio on ImageNet Dataset

| Model | Training Method | Top-1 Accuracy | Top-5 Accuracy | Gradient Size | Compression Ratio |
|---|---|---|---|---|---|
| AlexNet | Baseline | 58.17% | 80.19% | 232.56 MB | $1 \times$ |
| | TernGrad (Wen et al., 2017) | 57.28% (-0.89%) | 80.23% (+0.04%) | 29.18 MB [1] | $8 \times$ |
| | Deep Gradient Compression | **58.20%** (+0.03%) | **80.20%** (+0.01%) | **0.39 MB** [2] | **597** $\times$ |
| ResNet-50 | Baseline | 75.96 | 92.91% | 97.49 MB | $1 \times$ |
| | Deep Gradient Compression | **76.15** (+0.19%) | **92.97%** (+0.06%) | **0.35 MB** | **277** $\times$ |

**Speech Recognition**  The AN4 dataset contains 948 training and 130 test utterances (Acero, 1990) while Librispeech corpus contains 960 hours of reading speech (Panayotov et al., 2015). We use DeepSpeech architecture without n-gram language model, which is a multi-layer RNN following a stack of convolution layers (Hannun et al., 2014). We train a 5-layer LSTM of 800 hidden units per layer for AN4, and a 7-layer GRU of 1200 hidden units per layer for LibriSpeech, with *Nesterov momentum SGD* and gradient clipping, while learning rate anneals every epoch. The warm-up period for DGC is 1 epoch out of 80 epochs.

## 4.2 RESULTS AND ANALYSIS

We first examine Deep Gradient Compression on image classification task. Figure 3(a) and 3(b) are the Top-1 accuracy and training loss of ResNet-110 on Cifar10 with 4 nodes. The gradient sparsity is 99.9% (only 0.1% is non-zero). The learning curve of Gradient Dropping (Aji & Heafield, 2017) (red) is worse than the baseline due to gradient staleness. With momentum correction (yellow), the learning curve converges slightly faster, and the accuracy is much closer to the baseline. With momentum factor masking and warm-up training techniques (blue), gradient staleness is eliminated, and the learning curve closely follows the baseline. Table 2 shows the detailed accuracy. The accuracy of ResNet-110 is fully maintained while using Deep Gradient Compression.

When scaling to the large-scale dataset, Figure 3(c) and 3(d) show the learning curve of ResNet-50 when the gradient sparsity is 99.9%. The accuracy fully matches the baseline. An interesting observation is that the top-1 error of training with sparse gradients decreases faster than the baseline with the same training loss. Table 3 shows the results of AlexNet and ResNet-50 training on ImageNet with 4 nodes. We compare the gradient compression ratio with Terngrad (Wen et al., 2017) on AlexNet (ResNet is not studied in Wen et al. (2017)). Deep Gradient Compression gives 75×

---

[2]The gradient of the last fully-connected layer of Alexnet is 32-bit float. (Wen et al., 2017)

[2]We only transmit 32-bit values of non-zeros and 16-bit run lengths of zeros in flattened gradients.

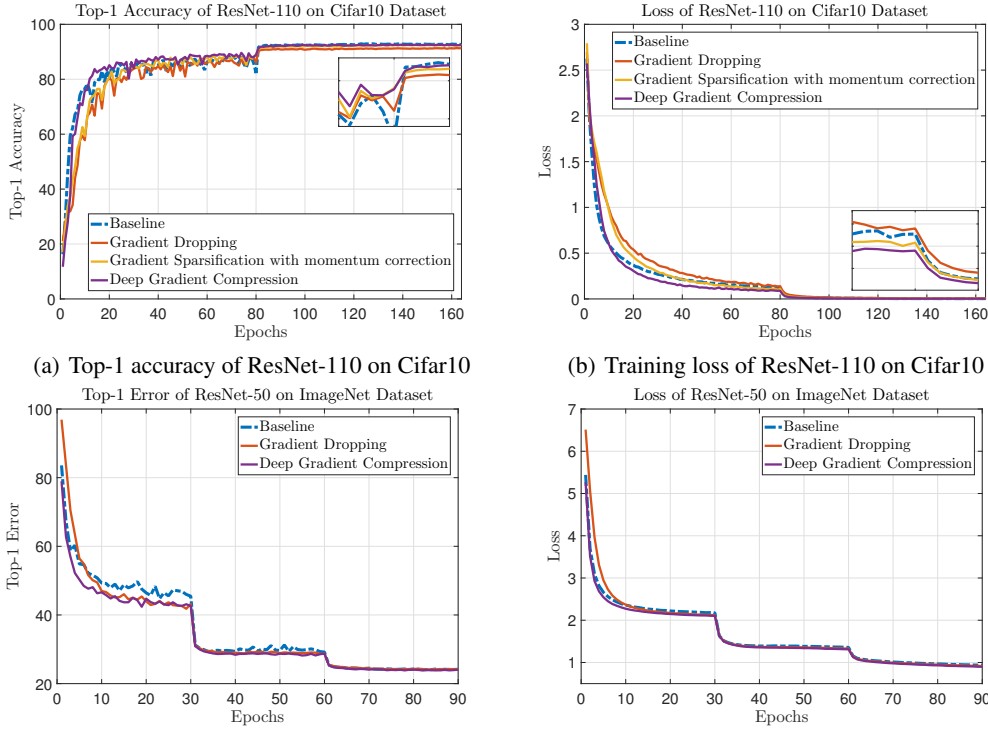

Figure 3: Learning curves of ResNet in image classification task (the gradient sparsity is 99.9%).

Table 4: Training results of language modeling and speech recognition with 4 nodes

| Task | Language Modeling on PTB | | | Speech Recognition on LibriSpeech | | | |
|---|---|---|---|---|---|---|---|
| Training Method | Perplexity | Gradient Size | Compression Ratio | Word Error Rate (WER) | | Gradient Size | Compression Ratio |
| | | | | test-clean | test-other | | |
| Baseline | 72.30 | 194.68 MB | 1 $\times$ | 9.45% | 27.07% | 488.08 MB | 1 $\times$ |
| Deep Gradient Compression | **72.24** **(-0.06)** | **0.42 MB** | **462** $\times$ | **9.06%** **(-0.39%)** | **27.04%** **(-0.03%)** | **0.74 MB** | **608** $\times$ |

better compression than Terngrad with no loss of accuracy. For ResNet-50, the compression ratio is slightly lower (277$\times$ vs. 597$\times$) with a slight increase in accuracy.

For language modeling, Figure 4 shows the perplexity and training loss of the language model trained with 4 nodes when the gradient sparsity is 99.9%. The training loss with Deep Gradient Compression closely match the baseline, so does the validation perplexity. From Table 4, Deep Gradient Compression compresses the gradient by 462 $\times$ with a slight reduction in perplexity.

For speech recognition, Figure 5 shows the word error rate (WER) and training loss curve of 5-layer LSTM on AN4 Dataset with 4 nodes when the gradient sparsity is 99.9%. The learning curves show the same improvement acquired from techniques in Deep Gradient Compression as for the image network. Table 4 shows word error rate (WER) performance on LibriSpeech test dataset, where *test-clean* contains clean speech and *test-other* noisy speech. The model trained with Deep Gradient Compression gains better recognition ability on both clean and noisy speech, even when gradients size is compressed by 608$\times$.

## 5 SYSTEM ANALYSIS AND PERFORMANCE

Implementing DGC requires gradient top-$k$ selection. Given the target sparsity ratio of 99.9%, we need to pick the top 0.1% largest over millions of weights. Its complexity is $O(n)$, where $n$ is the

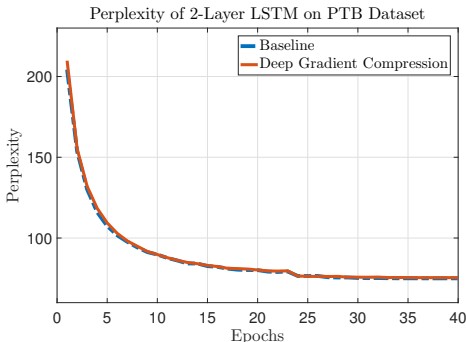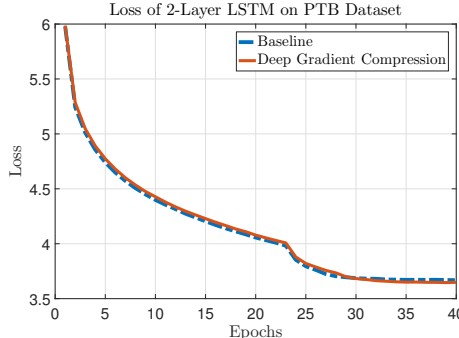

Figure 4: Perplexity and training loss of LSTM language model on PTB dataset (the gradient sparsity is 99.9%).

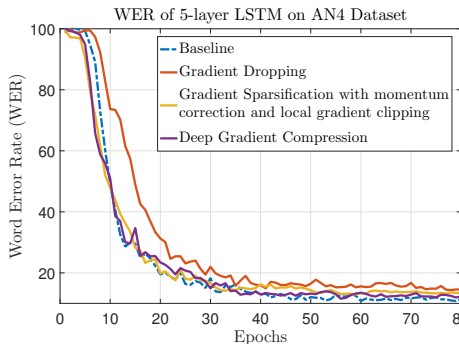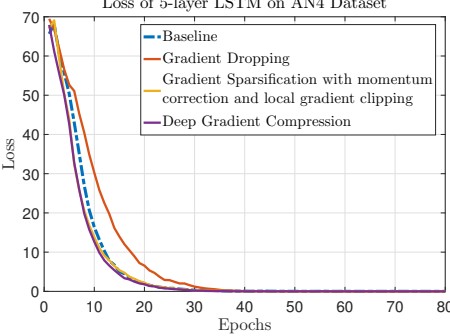

Figure 5: WER and training loss of 5-layer LSTM on AN4 (the gradient sparsity is 99.9%).

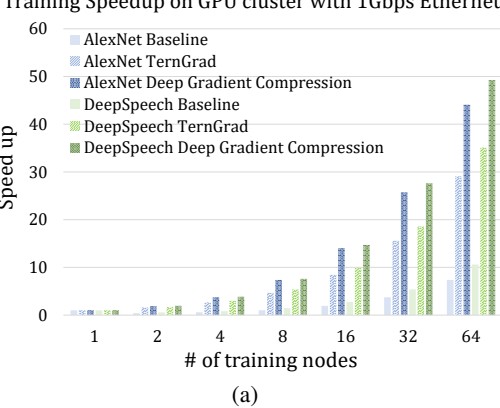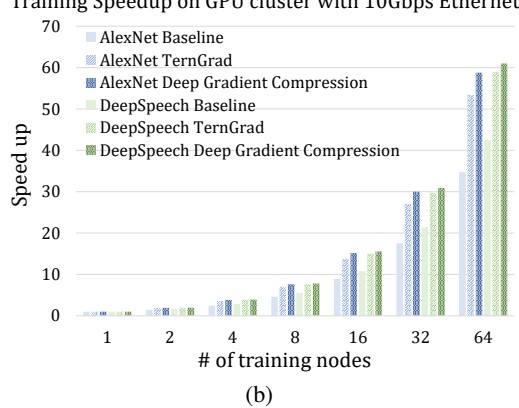

Figure 6: Deep Gradient Compression improves the speedup and scalability of distributed training. Each training node has 4 NVIDIA Titan XP GPUs and one PCI switch.

number of the gradient elements (Cormen, 2009). We propose to use sampling to reduce top-$k$ selection time. We sample only 0.1% to 1% of the gradients and perform top-$k$ selection on the samples to estimate the threshold for the entire population. If the number of gradients exceeding the threshold is far more than expected, a precise threshold is calculated from the already-selected gradients. Hierarchically calculating the threshold significantly reduces top-$k$ selection time. In practice, total extra computation time is negligible compared to network communication time which is usually from hundreds of milliseconds to several seconds depending on the network bandwidth.

We use the performance model proposed in Wen et al. (2017) to perform the scalability analysis, combining the lightweight profiling on single training node with the analytical communication modeling. With the all-reduce communication model (Rabenseifner, 2004; Bruck et al., 1997), the density of sparse data doubles at every aggregation step in the worst case. However, even considering this effect, Deep Gradient Compression still significantly reduces the network communication time, as implied in Figure 6.

Figure 6 shows the speedup of multi-node training compared with single-node training. Conventional training achieves much worse speedup with 1Gbps (Figure 6(a)) than 10Gbps Ethernet (Figure 6(b)). Nonetheless, Deep Gradient Compression enables the training with 1Gbps Ethernet to be competitive with conventional training with 10Gbps Ethernet. For instance, when training AlexNet with 64 nodes, conventional training only achieves about $30\times$ speedup with 10Gbps Ethernet (Apache, 2016), while with DGC, more than $40\times$ speedup is achieved with only 1Gbps Ethernet. From the comparison of Figure 6(a) and 6(b), Deep Gradient Compression benefits even more when the communication-to-computation ratio of the model is higher and the network bandwidth is lower.

## 6 CONCLUSION

Deep Gradient Compression (DGC) compresses the gradient by 270-600$\times$ for a wide range of CNNs and RNNs. To achieve this compression without slowing down the convergence, DGC employs momentum correction, local gradient clipping, momentum factor masking and warm-up training. We further propose hierarchical threshold selection to speed up the gradient sparsification process. Deep Gradient Compression reduces the required communication bandwidth and improves the scalability of distributed training with inexpensive, commodity networking infrastructure.

## 7 ACKNOWLEDGEMENT

This work was supported by National Natural Science Foundation of China (No.61622403), Tsinghua University Initiative Scientific Research Program and Beijing National Research Center for Information Science and Technology. We are also thankful to reviewers for their helpful suggestions.

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

## A    SYNCHRONOUS DISTRIBUTED STOCHASTIC GRADIENT DESCENT

In practice, each training node performs the forward-backward pass on different batches sampled from the training dataset with the same network model. The gradients from all nodes are summed up to optimize their models. By this synchronization step, models on different nodes are always the same during the training. The aggregation step can be achieved in two ways. One method is using the *parameter servers* as the intermediary which store the parameters among several servers (Dean et al., 2012). The nodes push the gradients to the servers while the servers are waiting for the gradients from all nodes. Once all gradients are sent, the servers update the parameters, and then all nodes pull the latest parameters from the servers. The other method is to perform the *All-reduce* operation on the gradients among all nodes and to update the parameters on each node independently (Goyal et al., 2017), as shown in Algorithm 2 and Figure 7. In this paper, we adopt the latter approach by default.

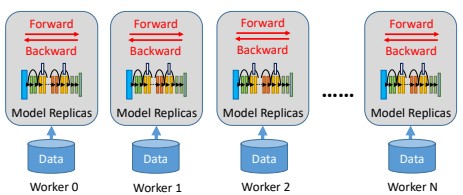

(a) Each node independently calculates gradients

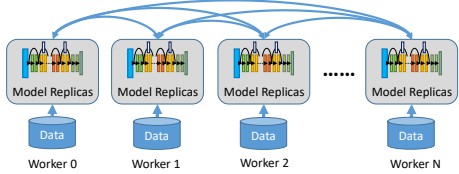

(b) All-reduce operation of gradient aggregation

Figure 7: Distributed Synchronous SGD

---

**Algorithm 2** Distributed Synchronous SGD on node $k$

**Input:** Dataset $\chi$
**Input:** minibatch size $b$ per node
**Input:** the number of nodes $N$
**Input:** Optimization Function $SGD$
**Input:** Init parameters $w = \{w[0], \cdots, w[M]\}$
1: **for** $t = 0, 1, \cdots$ **do**
2:     $G_t^k \leftarrow 0$
3:     **for** $i = 1, \cdots, B$ **do**
4:         Sample data $x$ from $\chi$
5:         $G_t^k \leftarrow G_t^k + \frac{1}{Nb} \nabla f(x; w_t)$
6:     **end for**
7:     *All-reduce* $G_t^k : G_t \leftarrow \sum_{k=1}^N G_t^k$
8:     $w_{t+1} \leftarrow SGD(w_t, G_t)$
9: **end for**

---

## B    GRADIENT SPARSIFICATION WITH NESTROV MOMENTUM CORRECTION

The conventional update rule for Nesterov momentum SGD (Nesterov, 1983) follows,

$$u_{t+1} = mu_t + \sum_{k=1}^N (\nabla_{k,t}), \quad w_{t+1} = w_t - \eta(m \cdot u_{t+1} + \nabla_t) \tag{8}$$

where $m$ is the momentum, $N$ is the number of training nodes, and $\nabla_{k,t} = \frac{1}{Nb} \sum_{x \in \mathcal{B}_{k,t}} \nabla f(x, w_t)$.

Before momentum correction, the sparse update follows,

$$v_{k,t+1} = v_{k,t} + \nabla_{k,t}, \quad u_{t+1} = mu_t + \sum_{k=1}^N sparse(v_{k,t+1}), \quad w_{t+1} = w_t - \eta u_{t+1} \tag{9}$$

After momentum correction sharing the same methodology with Equation 7, it becomes,

$$u_{k,t+1} = mu_{k,t} + \nabla_{k,t}, \quad v_{k,t+1} = v_{k,t} + (m \cdot u_{k,t+1} + \nabla_{k,t}), \quad w_{t+1} = w_t - \eta \sum_{k=1}^N sparse(v_{k,t+1}) \tag{10}$$

## C    LOCAL GRADIENT CLIPPING

When training the recurrent neural network with gradient clipping, we perform the gradient clipping locally *before* adding the current gradient $G_t^k$ to previous accumulation $G_{t-1}^k$ in Algorithm 1. Denote the origin threshold for the gradients L2-norm $||G||_2$ as $thr_G$, and the threshold for the local gradients L2-norm $||G^k||_2$ as as $thr_{G^k}$.

Assuming all $N$ training nodes have independent identical gradient distributions with the variance $\sigma^2$, the sum of gradients from all nodes have the variance $N\sigma^2$. Therefore,

$$E\left[||G^k||_2\right] \approx \sigma, \quad E\left[||G||_2\right] \approx N^{1/2}\sigma \tag{11}$$

Thus, We scale the threshold by $N^{-1/2}$, the current node's fraction of the global threshold,

$$thr_{G^k} = N^{-1/2} \cdot thr_G \tag{12}$$

## D  DEEP GRADIENT COMPRESSION ALGORITHM

| **Algorithm 3** Deep Gradient Compression for vanilla momentum SGD on node $k$ | **Algorithm 4** Deep Gradient Compression for Nesterov momentum SGD on node $k$ |
|---|---|
| **Input:** dataset $\chi$ | **Input:** dataset $\chi$ |
| **Input:** minibatch size $b$ per node | **Input:** minibatch size $b$ per node |
| **Input:** momentum $m$ | **Input:** momentum $m$ |
| **Input:** the number of nodes $N$ | **Input:** the number of nodes $N$ |
| **Input:** optimization function *SGD* | **Input:** optimization function *SGD* |
| **Input:** initial parameters $w = \{w[0], \cdots, w[M]\}$ | **Input:** initial parameters $w = \{w[0], \cdots, w[M]\}$ |
| 1: $U^k \leftarrow 0, V^k \leftarrow 0$ | 1: $U^k \leftarrow 0, V^k \leftarrow 0$ |
| 2: **for** $t = 0, 1, \cdots$ **do** | 2: **for** $t = 0, 1, \cdots$ **do** |
| 3: $\quad G_t^k \leftarrow 0$ | 3: $\quad G^k \leftarrow 0$ |
| 4: $\quad$ **for** $i = 1, \cdots, b$ **do** | 4: $\quad$ **for** $i = 1, \cdots, b$ **do** |
| 5: $\quad\quad$ Sample data $x$ from $\chi$ | 5: $\quad\quad$ Sample data $x$ from $\chi$ |
| 6: $\quad\quad G_t^k \leftarrow G_t^k + \frac{1}{Nb}\nabla f(x; \theta_t)$ | 6: $\quad\quad G_t^k \leftarrow G_t^k + \frac{1}{Nb}\nabla f(x; \theta_t)$ |
| 7: $\quad$ **end for** | 7: $\quad$ **end for** |
| 8: $\quad$ **if** Gradient Clipping **then** | 8: $\quad$ **if** Gradient Clipping **then** |
| 9: $\quad\quad G_t^k \leftarrow Local\_Gradient\_Clipping(G_t^k)$ | 9: $\quad\quad G_t^k \leftarrow Local\_Gradient\_Clipping(G_t^k)$ |
| 10: $\quad$ **end if** | 10: $\quad$ **end if** |
| 11: $\quad U_t^k \leftarrow m \cdot U_{t-1}^k + G_t^k$ | 11: $\quad U_t^k \leftarrow m \cdot \left(U_{t-1}^k + G_t^k\right)$ |
| 12: $\quad V_t^k \leftarrow V_{t-1}^k + U_t^k$ | 12: $\quad V_t^k \leftarrow V_{t-1}^k + U_t^k + G_t^k$ |
| 13: $\quad$ **for** $j = 0, \cdots, M$ **do** | 13: $\quad$ **for** $j = 0, \cdots, M$ **do** |
| 14: $\quad\quad thr \leftarrow s\%$ of $\left|V_t^k[j]\right|$ | 14: $\quad\quad thr \leftarrow s\%$ of $\left|V_t^k[j]\right|$ |
| 15: $\quad\quad Mask \leftarrow \left|V_t^k[j]\right| > thr$ | 15: $\quad\quad Mask \leftarrow \left|V_t^k[j]\right| > thr$ |
| 16: $\quad\quad \widetilde{G}_t^k[j] \leftarrow V_t^k[j] \odot Mask$ | 16: $\quad\quad \widetilde{G}_t^k[j] \leftarrow V_t^k[j] \odot Mask$ |
| 17: $\quad\quad V_t^k[j] \leftarrow V_t^k[j] \odot \neg Mask$ | 17: $\quad\quad V_t^k[j] \leftarrow V_t^k[j] \odot \neg Mask$ |
| 18: $\quad\quad U_t^k[j] \leftarrow U_t^k[j] \odot \neg Mask$ | 18: $\quad\quad U_t^k[j] \leftarrow U_t^k[j] \odot \neg Mask$ |
| 19: $\quad$ **end for** | 19: $\quad$ **end for** |
| 20: $\quad$ *All-reduce*: $G_t \leftarrow \sum_{k=1}^N encode(\widetilde{G}_t^k)$ | 20: $\quad$ *All-reduce*: $G_t \leftarrow \sum_{k=1}^N encode(\widetilde{G}_t^k)$ |
| 21: $\quad \theta_{t+1} \leftarrow SGD(\theta_t, G_t)$ | 21: $\quad \theta_{t+1} \leftarrow SGD(\theta_t, G_t)$ |
| 22: **end for** | 22: **end for** |

