# OpenReview forum: "Deep Gradient Compression: Reducing the Communication Bandwidth for Distributed Training"
_ICLR.cc/2018/Conference — Accept (Poster)_

### Official Review · AnonReviewer2 · 2017-11-23
**A useful contribution**

**Rating:** 7
**Confidence:** 4

**Review:**

I think this is a good work that I am sure will have some influence in the near future. I think it should be accepted and my comments are mostly suggestions for improvement or requests for additional information that would be interesting to have.

Generally, my feeling is that this work is a little bit too dense, and would like to encourage the authors in this case to make use of the non-strict ICLR page limit, or move some details to appendix and focus more on more thorough explanations. With increased clarity, I think my rating (7) would be higher.

Several Figures and Tables are never referenced in the text, making it a little harder to properly follow text. Pointing to them from appropriate places would improve clarity I think.

Algorithm 1 line 14: You never seem to explain what is sparse(G). Sec 3.1: What is it exactly that gets communicated? How do you later calculate the Compression Ratio? This should surely be explained somewhere.

Sec 3.2 you mention 1% loss of accuracy. A pointer here would be good, at that point it is not clear if it is in your work later, or in another paper. The efficient momentum correction is great!

As I was reading the paper, I got to the experiments and realized I still don't understand what is it that you refer to as "deep gradient compression". Pointer to Table 1 at the end of Sec 3 would probably be ideal along with some summary comments.

I feel the presentation of experimental results is somewhat disorganized. It is not clear what is immediately clear what is the baseline, that should be somewhere stressed. I find it really confusing why you sometimes compare against Gradient Dropping, sometimes against TernGrad, sometimes against neither, sometimes include Gradient Sparsification with momentum correction (not clear again what is the difference from DGC). I recommend reorganizing this and make it more consistent for sake of clarity. Perhaps show here only some highlights, and point to more in the Appendix.

Sec 5: Here I feel would be good to comment on several other things not mentioned earlier.
Why do you only work with 99.9% sparsity? Does 99% with 64 training nodes lead to almost dense total updates, making it inefficient in your communication model? If yes, does that suggest a scaling limit in terms of number of training nodes? If not, how important is the 99.9% sparsity if you care about communication cost dominating the total runtime? I would really like to better understand how does this change and what is the point beyond which more sparsity is not practically useful. Put differently, is DGC with 600x size reduction in total runtime any better than DGC with 60x reduction?


Finally, a side remark:
Under eq. (2) you point to something that I think could be more discussed. When you say what you do has the effect of increasing stepsize, why don't you just increase the stepsize?
There has recently been this works on training ImageNet in 1 hour, then in 24 minutes, latest in 15 minutes... You cite the former, but highlight different part of their work. Broader idea is that this is trend that potentially makes this kind of work less relevant. While I don't think that makes your work bad or misplaced, I think mentioning this would be useful as an alternative approach to the problems you mention in the introduction and use to motivate your contribution.
...what would be your reason for using DGC as opposed to just increasing the batch size?

---

> ### Author Response · Authors · 2017-12-13
> **Re: A useful contribution**
>
> We thank the reviewer for the comments.
>
>     -    Several Figures and Tables are never referenced in the text, making it a little harder to properly follow text. Pointing to them from appropriate places would improve clarity I think.
>
> We revised our paper. All the figures and tables are referenced properly in the text.
>
>     -    Algorithm 1 line 14: You never seem to explain what is sparse(G). Sec 3.1: What is it exactly that gets communicated? How do you later calculate the Compression Ratio?
>
> We have change the name of function to encode(G). The encode() function packs 32-bit nonzero gradient values and 16-bit run lengths of zeros in the flattened gradients. The encoded sparse gradients get communicated. These are described in the Sec 3.1 now.
> The compression ratio is calculated as follows:
>        The Gradient Compression Ratio = Size[ encode( sparse( G_k ) ) ] / Size [G_k]
> It is defined in the Sec 4.1 now.
>
>     -    Sec 3.2 you mention 1% loss of accuracy. A pointer here would be good, at that point it is not clear if it is in your work later, or in another paper.
>
> We pointed to the Figure 3(a) in the updated draft, and also cite the paper AdaComp [1].
>
>     -    Pointer to Table 1 at the end of Sec 3 would probably be ideal along with some summary comments.
>
> We make a summary at the end of Sec 3 and add Appendix D to show the overall algorithm of DGC in the updated draft.
>
>     -    I find it really confusing why you sometimes compare against Gradient Dropping, sometimes against TernGrad, sometimes against neither, sometimes include Gradient Sparsification with momentum correction (not clear again what is the difference from DGC).
>
> Because related work didn’t cover them all.  Gradient Dropping [2] only performed experiments on 2-layer LSTM for NMT, and 3-layer DNN for MNIST; TernGrad [3] only performed experiments on AlexNet, GoogleNet and VGGNet.  Therefore, we compared our AlexNet result with TernGrad.
>
> DGC contains not only momentum correction but also momentum factor masking and warm-up training. Momentum correction and Local gradient clipping are proposed to improve local gradient accumulation. Momentum factor masking and warm-up training are proposed to overcome the staleness effect. Comparison between Gradient Sparsification with momentum correction and DGC shows their impact on training respectively.
>
>     -    Why do you only work with 99.9% sparsity? Does 99% with 64 training nodes lead to almost dense total updates, making it inefficient in your communication model? If yes, does that suggest a scaling limit in terms of number of training nodes? If not, how important is the 99.9% sparsity if you care about communication cost dominating the total runtime?
>
> Yes, 99% with 128 training nodes lead to almost dense total updates, making it inefficient in communication. The scaling limit N in terms of number of training nodes depends on the gradient sparsity s: N ≈1/(1-s). When the gradient sparsity is 99.9%, the scaling limit is 1024 training nodes.
>
>     -    When you say what you do has the effect of increasing stepsize, why don't you just increase the stepsize? What would be your reason for using DGC as opposed to just increasing the batch size?
>
> Since the memory on GPU is limited, the way to increase the stepsize is to increase training nodes. Previous work in increasing the stepsize focus on how to deal with very large mini-batch training, while our work focus on how to reduce the communication consumption among increased nodes under poor network bandwidth. DGC can be considered as increasing the stepsize temporally on top of increasing the actual stepsize spatially.
>
> References:
> [1] Chen, Chia-Yu, et al. "AdaComp: Adaptive Residual Gradient Compression for Data-Parallel Distributed Training." arXiv preprint arXiv:1712.02679 (2017).
> [2] Aji, Alham Fikri, and Kenneth Heafield. Sparse Communication for Distributed Gradient Descent. In Empirical Methods in Natural Language Processing (EMNLP), 2017.
> [3] Wen, Wei, et al. TernGrad: Ternary Gradients to Reduce Communication in Distributed Deep Learning. In Advances in Neural Information Processing Systems, 2017.

---

> > ### Comment · AnonReviewer2 · 2018-01-02
> > **Re**
> >
> > Thanks for the response, I feel confident this contribution should be accepted.

---

### Official Review · AnonReviewer3 · 2017-11-27
**good empirical results, but requires work to justify the proposed techniques**

**Rating:** 6
**Confidence:** 5

**Review:**

This paper proposes additional improvement over gradient dropping(Aji & Heafield) to improve communication efficiency.

- First of all, the experimental results are thorough and seem to suggest the advantage of the proposed techniques.
- The result for gradient dropping(Aji & Heafield) should be included in the ImageNet experiment.
- I am having a hard time understanding the intuition behind v_t introduced in the momentum correction. The authors should provide some form of justifications.
   - For example, provide an equivalence provide to the original update rule or some error analysis would be great
   - Did you keep a running sum of v_t overall history? Such sum without damping(the m term in momentum update) is likely lead to the growing dominance of noise and divergence.
- The momentum masking technique seems to correspond to stop momentum when a gradient is synchronized. A discussion about the relation to asynchronous update is helpful.
- Do you do non-sparse global synchronization of momentum term? It seems that local update of momentum is likely going to diverge,  and the momentum masking somehow reset that.
- In the experiment, did you perform local aggregations of gradients between GPU cards before send out to do all0reduce in a network? since doing so will reduce bandwidth requirement.

In general, this is a paper shows good empirical results. But requires more work to justify the proposed correction techniques.


---

I have read the authors updates and changed my score accordingly(see series of discussions)

---

> ### Comment · AnonReviewer3 · 2017-12-09
> **the author  really need to provide justification to the momentum correction**
>
> I take a look at the other reviews after they get online. While most of them give accept to the paper given the strong empirical results provided by the paper.
>
> However, the problem is mentioned by all the reviewers(it is unclear why momentum correction is needed and the intuition behind this). I would like to emphasize that the current rule seems will is likely lead to the growing dominance of noise and divergence (because of no damping).
>
> I strongly encourage the author to clarify this. We cannot simply accept a paper for great empirical result without justification of why the rule works (all the reviewers are confused by this point)

---

> > ### Author Response · Authors · 2017-12-13
> > **Re: good empirical results, but requires work to justify the proposed techniques**
> >
> > We thank the reviewer for the comments. We have revised our paper.
> >
> >     -    Did you keep a running sum of v_t overall history? Such sum without damping(the m term in momentum update) is likely lead to the growing dominance of noise and divergence.  Do you do non-sparse global synchronization of momentum term? It seems that local update of momentum is likely going to diverge, and the momentum masking somehow reset that.
> >
> > We already revised our paper, and described the momentum correction more precisely in Section 3.2.
> >
> > Basically, the momentum correction performs the momentum SGD without update locally, and accumulates the velocity u_t locally. The optimization performs SGD with v_t instead of momentum SGD with G_t after momentum correction.  We add figure 2 to illustrate the difference.
> >
> > Therefore, we do not keep a running sum of v_t overall history, but keep a running sum of u_t. v_t is the running sum result and will be cleared after update (with or without momentum factor masking).  For example, at iteration t-1,
> > u_{t-1}  = m^{t-2} g_{1}+ … + m g_{t-2} + g_{t-1},
> > v_{t-1}  = (1+…+m^{t-2}) g_{1} + … + (1+m) g_{t-2} + g_{t-1}.
> > Update, w_{t} = w_{1} – lr x v_{t-1}
> > After update, v_{t-1}  = 0.
> > Next iteration,
> > u_{t} = m^{t-1} g_{1} + … + m g_{t-1} + g_{t},
> > v_{t}  = m^{t-1} g_{1} + … + m g_{t-1} + g_{t}.
> > Update, w_{t+1} = w_{t} – lr x v_{t}
> >                               = w_{1} – lr x (v_{t-1} + v_{t} )
> >                               = w_{1} - lr x [ (1+…+m^{t-1}) g_{1} + … + (1+m) g_{t-1} + g_{t} ]
> > Which is the same as the dense momentum SGD.
> >
> >     -    Did you perform local aggregations of gradients between GPU cards before send out to do all0reduce in a network?
> >
> > Yes.

---

> > > ### Comment · AnonReviewer3 · 2017-12-13
> > > **more improvements can be done on the momentum correction justification**
> > >
> > > This is the comment after author's updated a revised version of the paper.
> > >
> > > I now understand the proposed momentum correction method(with some effort). However, the presentation can be further improved.  I list my suggestions here:
> > >
> > > Clarify the approach:
> > > - The author should emphasize that the sparsification + thresholding have async nature, as the update only trigger when sparse condition applies and causes the mismatch problem.
> > > - It is a bit hard to understand the vanilla baseline(I might not do it in that way). It can be explained as local gradient aggregation and only apply momentum update at the trigger point. The update rule is no longer equivalent to SGD with momentum.
> > > - The paper did not explicitly mention that the value of v_t gets reset after triggering a communication, it should be explicitly mentioned in the update equation.
> > >
> > > Justify the correctness:
> > > Rough intuition gives most of the current justification. We can know that one is better than another, because of the equivalence.
> > > The author should try to do more formal justification, at least in some special cases, instead of leaving it as a debt for yourself or the potential readers
> > >
> > > - For example, the author should be able to prove that, under only one machine and one weight value. The weight value after K updates using the vanilla approach vs. the new approach.
> > > -A fundamental question that needs to be answered, is that why thresholding trigger method(which is async) works as good as sync SGD. A proof sketch to show the loss change after K step update would shed light on this and may give insights on what a good update rule is(possibly borrow some analysis from aync update and stale gradient)

---

> > > > ### Author Response · Authors · 2018-01-06
> > > > **Re: more improvements can be done on the momentum correction justification**
> > > >
> > > > We thank the reviewer for the suggestions. We have revised our paper.
> > > >
> > > > - The author should emphasize that the sparsification + thresholding have async nature, as the update only trigger when sparse condition applies and causes the mismatch problem.
> > > > We talked about the asynchrony nature in another way in the Sec 3.3: “Because we delay the update of small gradients, when these updates do occur, they are outdated or \emph{stale}.”
> > > >
> > > > - It is a bit hard to understand the vanilla baseline(I might not do it in that way). It can be explained as local gradient aggregation and only apply momentum update at the trigger point. The update rule is no longer equivalent to SGD with momentum.
> > > > We revised our paper to describe the vanilla sparse SGD + momentum baseline in Sec 3.2: "If SGD with the momentum is directly applied to the sparse gradient scenario (line 15 in Algorithm \ref{alg:ssgd}), the update rule is no longer equivalent to Equation \ref{eq:msgd}, which becomes:".
> > > >
> > > > - The paper did not explicitly mention that the value of v_t gets reset after triggering a communication, it should be explicitly mentioned in the update equation. Justify the correctness:
> > > > We revised our paper to explicitly mention resetting the value of v_t by the mask in Sec 3.2: "Similarly to the line 12 in Algorithm \ref{alg:ssgd}, the accumulation result $v_{k,t}$ gets cleared by the mask in the $sparse\left( \right)$ function."
> > > >
> > > > Rough intuition gives most of the current justification. We can know that one is better than another, because of the equivalence. The author should try to do more formal justification, at least in some special cases, instead of leaving it as a debt for yourself or the potential readers
> > > > - For example, the author should be able to prove that, under only one machine and one weight value. The weight value after K updates using the vanilla approach vs. the new approach.
> > > > We have shown that the sparsification + local gradient accumulation can be considered as “increasing the batch size over time” in Sec 3.1.
> > > >
> > > > -A fundamental question that needs to be answered, is that why thresholding trigger method(which is async) works as good as sync SGD. A proof sketch to show the loss change after K step update would shed light on this and may give insights on what a good update rule is(possibly borrow some analysis from aync update and stale gradient)
> > > > [1] has shown that running stochastic gradient descent (SGD) in an asynchronous manner can be viewed as adding a momentum-like term to the SGD iteration and a smaller momentum coefficient can work as good as sync SGD. We introduced the momentum factor masking to dynamically reduce the momentum term in Sec 3.3. Meanwhile, the asynchrony nature in the sparsification + local gradient accumulation can be considered as “increasing the batch size over time” in Sec 3.1, and there’s numerous work showing that increasing the batch size is feasible.
> > > >
> > > > References
> > > > [1] Mitliagkas, I., Zhang, C., Hadjis, S., & Re, C. (2017). Asynchrony begets momentum, with an application to deep learning. In 54th Annual Allerton Conference on Communication, Control, and Computing, Allerton 2016.

---

> > > ### Public Comment · (anonymous) · 2017-12-15
> > > **requires more work to justify momentum factor masking**
> > >
> > > - Therefore, we do not keep a running sum of v_t overall history, but keep a running sum of u_t. v_t is the running sum result and will be cleared after update (with or without momentum factor masking).  For example, at iteration t-1,
> > > u_{t-1}  = m^{t-2} g_{1}+ … + m g_{t-2} + g_{t-1},
> > > v_{t-1}  = (1+…+m^{t-2}) g_{1} + … + (1+m) g_{t-2} + g_{t-1}.
> > > Update, w_{t} = w_{1} – lr x v_{t-1}
> > > After update, v_{t-1}  = 0.
> > > Next iteration,
> > > u_{t} = m^{t-1} g_{1} + … + m g_{t-1} + g_{t},
> > > v_{t}  = m^{t-1} g_{1} + … + m g_{t-1} + g_{t}.
> > > Update, w_{t+1} = w_{t} – lr x v_{t}
> > >                               = w_{1} – lr x (v_{t-1} + v_{t} )
> > >                               = w_{1} - lr x [ (1+…+m^{t-1}) g_{1} + … + (1+m) g_{t-1} + g_{t} ]
> > > Which is the same as the dense momentum SGD.
> > >
> > > In the momentum factor masking section, both v_t and u_t get reset after trigerring a communication. Why only v_t gets reset in your example?

---

> > > > ### Author Response · Authors · 2017-12-21
> > > > **Re: requires more work to justify momentum factor masking**
> > > >
> > > >         Thank you for your question.
> > > >         This example is to show how the momentum correction works, and therefore we do not consider the staleness effect in this example.
> > > >         Suppose the last update is at iteration t-1, the next update at iteration t+T-1, and we only consider the gradients  { g_{t}, g_{t+1}, ..., g_{t+T-1} }
> > > >          - Dense Update
> > > >             w_{t+T} = w_{t} - lr x [ ... + (1 + m +  ... + m^{T-1}) x g_{t} +  (1 + m +  ... + m^{T-2}) x g_{t+1} + ... + 1 x g_{t+T-1}]
> > > >             w_{t+\tau} = w_{t} - lr x [ ... + (1 + m +  ... + m^{\tau-1}) x g_{t} +  (1 + m +  ... + m^{\tau-2}) x g_{t+1} + ... + (1 + m +  ... + m^{\tau-T}) x g_{t+T-1} + ...], where \tau > T
> > > >
> > > >          - Only local gradient accumulation
> > > >             the coefficients of  { g_{t}, g_{t+1}, ..., g_{t+T-1} } are always the same.
> > > >             w_{t+T} = w_{t} - lr x [ ... + 1 x g_{t} +  1 x g_{t+1} + ... +  1 x g_{t+T-1}]
> > > >             w_{t+\tau} = w_{t} - lr x [ ... + (1 + m + m^2 + ... + m^{\tau-T}) x (g_{t} + g_{t+1} + ... + g_{t+T-1}) + ...]
> > > >
> > > >          - With the momentum correction,
> > > >             the coefficients of  { g_{t}, g_{t+1}, ..., g_{t+T-1} } are always the same as the dense update.
> > > >             w_{t+T} = w_{t} - lr x [ ... + (1 + m +  ... + m^{T-1}) x g_{t} +  (1 + m +  ... + m^{T-2}) x g_{t+1} + ... + 1 x g_{t+T-1}]
> > > >             w_{t+\tau} = w_{t} - lr x [ ... + (1 + m +  ... + m^{\tau-1}) x g_{t} +  (1 + m +  ... + m^{\tau-2}) x g_{t+1} + ... + (1 + m +  ... + m^{\tau-T}) x g_{t+T-1} + ...], where \tau > T
> > > >
> > > >          - With the momentum correction and momentum factor masking
> > > >             we clear the local u_{t} to prevent the delayed gradients from misleading the optimization after they are used for the update.
> > > >             w_{t+T} = w_{t} - lr x [ ... + (1 + m +  ... + m^{T-1}) x g_{t} +  (1 + m +  ... + m^{T-2}) x g_{t+1} + ... + 1 x g_{t+T-1}]
> > > >             w_{t+\tau} = w_{t} - lr x [ ... + (1 + m +  ... + m^{T-1}) x g_{t} +  (1 + m +  ... + m^{T-2}) x g_{t+1} + ... + 1 x g_{t+T-1} + ...], where \tau > T

---

### Official Review · AnonReviewer1 · 2017-11-29
**Study on gradient compression**

**Rating:** 7
**Confidence:** 4

**Review:**

The paper is thorough and on the whole clearly presented. However, I think it could be improved by giving the reader more of a road map w.r.t. the guiding principle. The methods proposed are heuristic in nature, and it's not clear what the guiding principle is. E.g., "momentum correction". What exactly is the problem without this correction? The authors describe it qualitatively, "When the gradient sparsity is high, the interval dramatically increases, and thus the significant momentum effect will harm the model performance". Can the issue be described more precisely? Similarly for gradient clipping, "The method proposed by Pascanu et al. (2013) rescales the gradients whenever the sum of their L2-norms exceeds a threshold. This step is conventionally executed after gradient aggregation from all nodes. Because we accumulate gradients over iterations on each node independently, we perform the gradient clipping locally before adding the current gradient... " What exactly is the issue here? It reads like a story of what the authors did, but it's not really clear why they did it.

The experiments seem quite thorough, with several methods being compared. What is the expected performance of the 1-bit SGD method proposed by Seide et al.?

re. page 2: What exactly is "layer normalization"?

re. page 4: What are "drastic gradients"?

---

> ### Author Response · Authors · 2017-12-13
> **Re: Study on gradient compression**
>
>        We thank the reviewer for the comments.
>
>     -    What exactly is the problem without this correction? Can the issue be described more precisely?
>
> We already revised our paper, and described the momentum correction more precisely in Section 3.2. Basically, the momentum correction performs the momentum SGD without update locally and accumulates the velocity u_t locally.
>
>     -    What exactly is the issue of Gradient clipping?
>
> When training RNN, people usually use Gradient Clipping to avoid the exploding gradient problem. The hyper-parameter for Gradient Clipping is the threshold thr_G of the gradients L2-norm. The gradients for optimization is scaled by a coefficient depending on their L2-norm.
>
> Because we accumulate gradients over iterations on each node independently, we need to scale the gradients before adding them to the previous accumulation, in order to scale the gradients by the correct coefficient. The threshold for local gradient clipping thr_Gk should be set to N^{-1/2} x thr_G. We add Appendix C to explain how N^{-1/2} comes.
>
>     -    What is the expected performance of the 1-bit SGD method proposed by Seide et al.?
>
> 1-bit SGD [1] encodes the gradients as 0 or 1, so the data volume is reduced by 32x. Meanwhile, since 1-bit SGD quantizes the gradients column-wise, a floating-point scaler per column is required, and thus it cannot yield much speed benefit on convolutional neural networks.
>
>     -    What exactly is "layer normalization"
>
> “Layer Normalization” is similar to batch normalization but computes the mean and variance from the summed inputs in a layer on a single training case. [2]
>
> -	 What are "drastic gradients"?
>
> It means the period when the network weight changes dramatically.
>
> References:
> [1] Frank Seide, Hao Fu, Jasha Droppo, Gang Li, and Dong Yu. 1-bit stochastic gradient descent and its application to data-parallel distributed training of speech DNNs. In Fifteenth Annual Conference of the International Speech Communication Association, 2014.
> [2] J. Lei Ba, J. R. Kiros, and G.E.Hinton, Layer Normalization. ArXiv e-prints, July 2016

---

### Public Comment · (anonymous) · 2017-11-14
**Does compression ratio consider the larger communication overhead in warm-up training and the smaller sparsity after summing?**

For compression ratio in Table 3 & 4, does this work consider the larger communication volume during warm-up training?
Please clarify how many iterations it took to "warm-up" and what was the sparsity of gradients during warm-up. If it did warm up for 20% of total epochs with 50% sparsity, the compression ratio is bounded by 10x.

Did this work consider a larger communication volume of summed gradients? Suppose there are gradients from k workers  to sum up and the sparsity of gradients is s, the expectation of the sparsity of summed gradients is s^n which exponentially decreases with n. Please clarify this.

Thanks

---

> ### Author Response · Authors · 2017-11-15
> **we have considered these non-ideal effects.**
>
> we thank the reviewer for the comments.
>
> (1)
> First, warm-up training takes only 4 out of 90 epochs for imagenet, 1 out of 70 epochs for librispeech, which is only 1.4%-4% of the total training epochs. Therefore the impact of warmup training is negligible.
>
> Second, during warm-up training the gradient is also very sparse. On Imagenet, the sparsity for the 4 warm-up epochs are: 75% -> 93.75% -> 98.4375% -> 99.6% (exponentially increase), then 99.9% for the rest 86 epochs. Same warm-up sparsity rule applies to the first four quarter epochs on Librispeech, then 99.9% for the rest 69 epochs.
>
>
> (2) Yes, we already considered the a larger communication volume of summed gradients.
>
> With N=2^k workers and s sparsity. We need k step to gather these gradients. The density doubles at every step, so the average communication volume is \sum_{i=0}^{k-1} 2^{i}*M*s/k = (N-1)/log(N)*M*s. The average density increases sub-linearly with the number of nodes by N/log(N), not exponentially.
>
> We already considered this non-ideal effect in the second paragraph of Section 5: "the density of sparse data doubles at every aggregation step in the worst case. However, even considering this effect, Deep Gradient Compression still significantly reduces the network communication time, as implied in Figure 6." "For instance, when training AlexNet with 64 nodes, conventional training only achieves nearly 30× speedup with 10Gbps Ethernet (Apache, 2016), while with DGC, more than 40× speedup is achieved even with 1Gbps Ethernet". With 1Gbps Ethernet, the speedup of TernGrad is 30x, our worse case is 44x (considering this non-ideal effect), our best case is 58x. We reported the worse case, which is 44x speedup (see Figure 6).

---

### Public Comment · ~Quan_Vuong2 · 2017-11-15
**Typos and clarification questions**

Thank you for a great paper! The author's intuition really shines through. I just have a few clarifying points:

- Equation 1 & 2: shouldn’t k start from 1 if N is the number of training node ?
- Related Work section: Graidient typo
- Section 4.2: csparsity typo
- Line 8, 9 in Algorithm 4 in Appendix B: shouldn’t line 8 be U <- mU + G and line 9 be V_t <- V_{t-1} + mU + G
- Is "Gradient Size" referring to the average size of the gradient that's larger than the threshold ?

edit 1: add question about gradient size.

---

> ### Author Response · Authors · 2017-11-15
> **Re: Typos and clarification questions**
>
> We really appreciate your comments.
>
> - Equation 1 & 2: shouldn’t k start from 1 if N is the number of training node?
>        Yes. It's a typo. k should start from 1.
>
> - Related Work section: Graidient typo
> - Section 4.2: csparsity typo
>        Thank you for pointing out these typos. They should be "Gradient" and "sparsity".
>
> - Line 8, 9 in Algorithm 4 in Appendix B: shouldn’t line 8 be U <- mU + G and line 9 be V_t <- V_{t-1} + mU + G
>        These two lines are equivalent to those in Algorithm 4 in Appendix B.
>
> - Is "Gradient Size" referring to the average size of the gradient that's larger than the threshold?
>        Yes. "Gradient Size" is referring to the size of the sparse gradient, which contains both the gradient values that are larger than the threshold and 16-bit index distances when it comes to DGC.

---

### Public Comment · ~Kenneth_Heafield1 · 2017-11-15
**Hello from your baseline**

I'm Kenneth Heafield, one of the authors cited.

It's an interesting bag of 4 tricks here and I will likely use them going forward.

"Gradient Dropping requires adding a layer normalization." Figure 5 in our paper shows that gradient dropping works, admittedly slower, without layer normalization if we determine the threshold locally to each parameter/matrix rather than globally.

I feel like you're giving us too much credit.  Strom https://s3-us-west-2.amazonaws.com/amazon.jobs-public-documents/strom_interspeech2015.pdf and Dryden et al https://ornlcda.github.io/MLHPC2016/papers/3_Dryden.pdf deserve to be cited too.

Warm-up training works in general, so was it included in your baseline experiments as well?

"incurring 0.3% loss of accuracy on a machine translation task" It would be better to say BLEU score here, rather than a vague metric.  Parsing people fight over 0.3% while translation people shrug over 0.3% BLEU.

"Implementing DGC requires gradient sorting."  To be pedantic, it requires top-k selection which we have been talking to NVIDIA about implementing more efficiently in the context of beam search.  I like the hierarchical add-on to the sampling we've been doing too; if too few gradients pass the threshold, do you sample more?

Abstracts should compare to the strongest baseline, not just the stock baseline.

Let's talk when you're less anonymous.

---

> ### Author Response · Authors · 2017-11-18
> **Re: Hello from your baseline**
>
> Dear Kenneth Heafield,
>
>      Thank you for clarifying the Gradient Dropping, it's very helpful. We will describe the Gradient Dropping in a more rigorous way in the final version.
>      We also appreciate your reminding us of citing these two excellent papers.
>      Here are the answers to your questions.
>
>      - Warm-up training works in general, so was it included in your baseline experiments as well?
>
>        Warm-up training was previously used for improving the large minibatch training proposed by Goyal et. al. They warm up the learning rate linearly in the first several epochs. However, we are the first to warm up the sparsity during the gradient pruning. Therefore, only experiments with DGC adopted warm-up sparsity. It is a simple but effective technique.
>
>       - "Implementing DGC requires gradient sorting."  To be pedantic, it requires top-k selection which we have been talking to NVIDIA about implementing more efficiently in the context of beam search.  I like the hierarchical add-on to the sampling we've been doing too; if too few gradients pass the threshold, do you sample more?
>
>       We indeed use top-k selection instead of sorting. We do not sample more if too few gradients are selected. Since hierarchical selection is designed to control the communication data size, we will perform top-k selection twice only when too many gradients pass the threshold.

---

### Public Comment · ~Wei_Wen1 · 2017-11-28
**Hi from TernGrad**

Hi from TernGrad,

Impressive result, really!

For the top-1 accuracy in Table 3, I guess the 0.89% accuracy difference of TernGrad comes from the different ways we trained the standard AlexNet? In our work, the baseline AlexNet is trained using the same hyper-parameters of caffe (https://github.com/BVLC/caffe/tree/master/models/bvlc_alexnet), and converges to 57.32%. Your baseline got 58.17% because you used different training hyper-parameters in Wilber 2016 as you pointed out?
Replacing floating SGD by TernGrad, it converges to 57.28%. The loss because of TernGrad is just 0.04% instead of 0.89%?

Is it easy to implement all of the techniques? Do you plan to open source it? I may want to try this.
The core of TernGrad can be done within several lines (https://github.com/wenwei202/terngrad/blob/master/terngrad/inception/bingrad_common.py#L159-L166).

And just be curious about how does the warmup stage generalize, does the same warmup scheme work in general for all experiments? I am asking since we may not want to tune the warmup stage for several times when training a DNN, which essentially is wasting training time. TernGrad converges with the same hyper-parameters of standard SGD.

Thanks,
-Wei

---

> ### Author Response · Authors · 2017-12-13
> **Re: Hi from TernGrad**
>
> Hi, Wei. Thank you for your comments.
>
>     First of all, all the hyper-parameters, including the learning rate and momentum, are the same as the default settings.
>
>     -    The loss because of TernGrad is just 0.04% instead of 0.89%?
>
>      In the paper of TernGrad [1], the baseline AlexNet is trained with dropout ratio of 0.5, while the TernGrad AlexNet is trained with dropout ratio of 0.2. The paper claims that quantization introduces randomness and less dropout ratio avoids over-randomness. However, when we trained the baseline AlexNet with dropout ratio of 0.2, we gained 1 point improvement in top-1 accuracy. It indicates that the TernGrad might incur more loss of accuracy than expected. Therefore, to be fair, we use the dropout ratio of 0.2 in all experiments relating to AlexNet.
>
>    -    does the same warmup scheme work in general for all experiments?
>
>    Yes. Warm-up training takes only 4 out of 90 epochs for ImageNet, 1 out of 70 epochs for Librispeech. The gradient sparsity increases exponentially  75% -> 93.75% -> 98.4375% -> 99.6%.
>
> References:
> [1] Wen, Wei, et al. TernGrad: Ternary Gradients to Reduce Communication in Distributed Deep Learning. In Advances in Neural Information Processing Systems, 2017.

---

### Public Comment · (anonymous) · 2017-12-05
**Some details on distributed training of language model with Deep Gradient Compression**


Hello!

Could you please tell what was the batch size and the number of iterations per epoch on a node during distributed training of the language model on PTB? This is necessary to get an idea of total amount of communication that was sufficient to reach perplexity 72.24 at the end of 40-th epoch.

Thank you!

---

> ### Author Response · Authors · 2017-12-13
> **Re: Some details on distributed training of language model with Deep Gradient Compression**
>
> Thank you for your comments.
> The batch size is 80 and the number of iterations per epoch is 332.

---

### Public Comment · (anonymous) · 2017-12-05
**details on implementation of Deep Gradient Compression**

In the appendix, you mention two ways to aggregate gradients: parameter server and All-reduce.
1.For parameter server, communication is reduce when push sparse gradient to parameter server. Is it possible to pull sparsified gradient and applied locally?
2. For All-reduce, since the sparse gradients may be of different size, the standard MPI All-reduce operation won't work for this. Do you implement  your own All-reduce operation?

---

> ### Author Response · Authors · 2017-12-13
> **Re: details on implementation of Deep Gradient Compression**
>
> Thank you for your comments.
>     -     For parameter server, communication is reduced when push sparse gradient to parameter server. Is it possible to pull sparsified gradient and applied locally?
>
>       Yes, you can pull sparsified gradient and applied locally.
>
>    -     For All-reduce, since the sparse gradients may be of different size, the standard MPI All-reduce operation won't work for this. Do you implement your own All-reduce operation?
>
>       In our experiments, we force the size of the sparse gradients to be same as 0.1% of the number of gradients. We use hierarchical top-k selection not only to speed up sparsification but also to control the sparse gradients size. If the number of gradients is smaller than 0.1%, we filled the buffer with zeros. If it is much larger, we re-calculate the threshold. However, an efficient All-reduce operation for sparse communication is one of our future work.

---

> > ### Public Comment · ~Dongxu_Wang1 · 2018-10-16
> > **how do you implement All-reduce in your paper?**
> >
> > Since you didn't implement 'sparse communication All-reduce', how do you implement All-reduce in your paper?
> > Could you please explain more details about your All-reduce implementation and the data structure of sparse gradient buffer? And, how do you implement encode() and decode()？  Thank you~

---

### Public Comment · (anonymous) · 2017-12-10
**some suggestions about deep gradient compression**

This paper has strong experimental results. Momentum and learning rate correction make sense for effective larger mini-batch size.  However there are some suggestions about this work.


1. This submission should cite other papers well.  The main algorithm of this submission is very similar as Dryden's work in 2016 ( Communication quantization for data-parallel training of deep neural networks) and Strom in 2015 (Strom,  N.   2015.   Scalable  distributed  dnn  training  using commodity gpu cloud computing. In Sixteenth Annual Conference of the International Speech Communication Association).  Moreover, recently an ArXiv paper (AdaComp : Adaptive Residual Gradient Compression for Data-Parallel Distributed Training, accepted in AAAI18) also reported similar gradient compression scheme and shows excellent experimental results in all different NNs (ResNet50, ResNet18, AlexNet, RNN, DNN, LSTM etc..).  This paper should cite relevant work properly.

2. In section5, authors proposed sampling to reduce sorting time (the same as Dryden's work in 2016).  Although sorting could be Nlog(N), this method is not easy to be parallel.  Thus, large computation overhead still exists.

3.  Learning rate correction (estimate T), momentum correction, momentum factor mask, and warm up are very empirical.  From previous works, gradient residue compression is pretty robust, it is not surprising that the compression rate is high.

4. The paper just focuses on compression from workers to parameter server.  What happened in the direction from parameter to workers?  This could reduce their compression rate by learner number (as described in TernGrad).

5. What is the sparse representation?  The overhead of sparse representation should be discussed.  It is easy to lose compression rate by >10x here.  The high compression rate may be confusing if detailed sparse representation is not discussed.

6. How much warm up period do you need to use for each examples? Warm-up makes the experiments much easier since they do not clearly mention the warm-up epoch number.

7.  Is the compression layer-wise or whole model (including FC layers and convolution layers)?

In general, this paper reused previous gradient residue idea and added momentum and learning rate correction for effective larger mini-batch size.  This paper did a lot of experiments and has strong experimental results for NN convergence.  However this paper added several hyper-parameters (momentum correction, learning rate correction, warm up, and momentum factor mask etc..) and should clearly list values of these parameters in the test cases.  It is also important to guide users ways to put these extra hyper-parameters.  The compression rate and performance ignore several important factors such as sparsity representation, different directions of compression, computation overhead (parallel or not); results are from simple model only.  Look forward to seeing more exciting papers from this team!

---

> ### Author Response · Authors · 2017-12-13
> **Re: Some suggestions about deep gradient compression**
>
> Thank you for your suggestions.
> We appreciate your reminding us of citing these excellent papers, and we have already cited these work in the newest version of our paper.
>
>     -    Although sorting could be Nlog(N), this method is not easy to be parallel.  Thus, large computation overhead still exists.
>
> We use top-k selection, *NOT* sorting. The complexity of top-k selection is O(N), not O(NlogN) [1]. To further reduce computation, we perform the top-k selection on samples in stride. The sample rate is 0.1% to 1%. In practice, without any code optimization, the extra computation takes less than 10% of total communication time when training AlexNet with 64 nodes under 1Gbps Ethernet. We have already included this in Figure 6.
>
>     -    From previous works, gradient residue compression is pretty robust, it is not surprising that the compression rate is high.
>
> In fact, gradient residue compression does not preserve  the accuracy of the model.
> Figure 4 in the related work [2] shows that gradient residue compression brings around 2% to 5% loss of accuracy when the compression ratio is less than 500x, and even damages the training when the compression ratio is higher. It is our bag of 4 techniques that enables no loss of accuracy.
>
>     -    What happened in the direction from parameter to workers?  This could reduce their compression rate by learner number.
>
> First, we use all-reduce communication model in system performance analysis.
>
> With N=2^k workers and s sparsity. We need k step to gather these gradients. The density doubles at every step, so the average communication volume is \sum_{i=0}^{k-1} 2^{i}*M*s/k = (N-1)/log(N)*M*s. The average density increases sub-linearly with the number of nodes by N/log(N), not exponentially.
>
> We already considered this non-ideal effect, including the extra computation cost on top-k selection, in the second paragraph of Section 5: "the density of sparse data doubles at every aggregation step in the worst case. However, even considering this effect, Deep Gradient Compression still significantly reduces the network communication time, as implied in Figure 6." "For instance, when training AlexNet with 64 nodes, conventional training only achieves nearly 30× speedup with 10Gbps Ethernet (Apache, 2016), while with DGC, more than 40× speedup is achieved even with 1Gbps Ethernet". With 1Gbps Ethernet, the speedup of TernGrad is 30x, our worse case is 44x (considering this non-ideal effect), our best case is 58x. We reported the worse case, which is 44x speedup (see Figure 6).
>
> When it comes to parameter server communication model, we only pull the sum of sparse gradients, which is the same as TernGrad [3]. With the gradient compression ratio of 500x, it requires at least 500 training nodes to pull the same data size as in the dense scenario.
>
>     -    What is the sparse representation?
>
> We already discussed the sparse representation strategy in section 3.1. We used the simple run-length encoding: we pack the 32-bit float nonzero gradient values and 16-bit run lengths of zeros of the flattened gradients. The overhead is only 0.5x, not 10x. We already considered the overhead when reporting the compression ratio.
>
>     -    How much warm up period do you need to use for each examples?
>
> Warm-up training takes only 4 out of 90 epochs for ImageNet, 1 out of 70 epochs for Librispeech, which is only 1.4%-4% of the total training epochs. The time impact of warmup training is negligible.
>
>     -    Is the compression layer-wise or whole model (including FC layers and convolution layers)?
>
> Unlike AdaComp [2] has “~200X for fully-connected and recurrent layers, and ~40X for convolutional layers”, our compression rate is the same for the WHOLE model, where sparsity=99.9% for ALL layers.

---

> > ### Author Response · Authors · 2017-12-13
> > **continue**
> >
> > (continue)
> >
> >     -    However this paper added several hyper-parameters (momentum correction, learning rate correction, warm up, and momentum factor mask etc..)
> >
> > The *only* hyper-parameters introduced by DGC are the warm-up training strategy. However, we use the same settings in all experiments as answered above. Momentum correction and Momentum factor masking are equation changes, they do not introduce any hyper-parameters.
> >
> >     -    The compression rate and performance ignore several important factors such as sparsity representation, different directions of compression, computation overhead (parallel or not)
> >
> > No, Figure 6 in the Sec 5 already takes the sparsity representation, computation overhead, communication overhead into account.
> >
> >     -    Results are from simple model only
> >
> > No, we have broadly experimented on state-of-the-art, complex models across CNN, RNN, CNN and RNN mixture. We extensively experimented with ResNet110 on Cifar10, AlexNet/ResNet50 on ImageNet, 2-layer LSTM with the size of 195MB on PTB, 7-layer GRU following 3-layer CNN (DeepSpeech) with the size of 488MB on LibriSpeech.
> > In comparison, previous work Gradient Dropping [4] performed experiments on 2-layer LSTM with size of 476MB for NMT, and 3-layer DNN with size of 80MB on MNIST;
> > TernGrad [3] performed experiments on AlexNet, GoogleNet, and VGGNet on ImageNet;
> > Adacomp [2] performed experiments on 4-layer CifarCNN with the size of 0.3MB on Cifar10, AlexNet, ResNet18, ResNet50 on ImageNet, BN50-DNN with the size of 43MB on BN50, and 2-layer LSTM with the size of 13MB on Shakespeare Dataset.
> >
> > References:
> > [1] Cormen, Thomas H. Introduction to algorithms. MIT press, 2009
> > [2] Chen, Chia-Yu, et al. "AdaComp: Adaptive Residual Gradient Compression for Data-Parallel Distributed Training." arXiv preprint arXiv:1712.02679 (2017).
> > [3] Wen, Wei, et al. TernGrad: Ternary Gradients to Reduce Communication in Distributed Deep Learning. In Advances in Neural Information Processing Systems, 2017.
> > [4] Aji, Alham Fikri, and Kenneth Heafield. Sparse Communication for Distributed Gradient Descent. In Empirical Methods in Natural Language Processing (EMNLP), 2017.

---

### Public Comment · ~Chia-Yu_Chen1 · 2017-12-13
**Hi from AdaComp**

Hi DeepGradientCompression,

This is an interesting paper.  I think that it is misleading to mention that AdaComp shows 0.2-0.4% degradation.  AdaComp sometimes actually shows ~0.3% improvement and it always <0.5% difference compared to baseline.  The difference is from randomness of SGD; not from AdaComp itself.  It is not very meaningful to quote SGD difference within 0.5%.  Please correct it.

By the way, I have some questions: how many workers do you use in the experiments?  What is worker-scalability of deep gradient compression?

Best,

Chia-Yu

---

> ### Author Response · Authors · 2017-12-14
> **Fully maintaining the accuracy on ImageNet at high compression ratio is not easy, but Deep Gradient Compression made it.**
>
> Thanks for the comments. The accuracy degradation on ImageNet are quoted from the Table 2 of AdaComp [1]:
> ResNet18: baseline top1 error=32.41%, AdaComp top1 error=32.87% (0.46% accuracy degradation)
> ResNet50: baseline top1 error=28.91%, AdaComp top1 error=29.15% (0.24% accuracy degradation)
>
> In our DGC work:
> ResNet50: baseline top1 error=24.04%, DGC top1 error=23.85%
>
> We respect your argument and would be happy to adjust the citation to your paper. However, we believe ImageNet results are more interesting than MNIST. The 0.5% Top1 accuracy degradation on ImageNet is significant, not noise. Fully maintaining the accuracy on ImageNet at a much higher compression ratio is not easy, while the bag of four techniques introduced in DGC achieved this.
>
> The worker-scalability of deep gradient compression is described in Figure6 with up to 64 workers.
>
> References:
> [1] Chen, Chia-Yu, et al. "AdaComp: Adaptive Residual Gradient Compression for Data-Parallel Distributed Training." arXiv preprint arXiv:1712.02679 (2017).

---

> > ### Public Comment · ~Chia-Yu_Chen1 · 2017-12-22
> > **Thanks for your response**
> >
> > Hi DGC,
> >
> > Thanks for your comments.  I read the paper again.  The idea is quite interesting, but I still cannot say that I totally understood the results.  Need more time for me to digest.
> >
> >
> > Sincerely,
> >
> > Chia-Yu

---

### Public Comment · (anonymous) · 2017-12-16
**question about Momentum Factor Masking**

Once we applied momentum factor masking; the momentum correction becomes useless.  The  accumulated discounting factor in Eq(6) was masked and become the same as original way.  It is not clear about momentum factor mask.  Please clarify it.  As reviewer's comments, it seems that momentum factor mask resets the momentum correction.  From the content, this is really confusing!  Thanks a lot.

---

> ### Author Response · Authors · 2017-12-21
> **Re: question about Momentum Factor Masking**
>
> Thank you for your comments.
>
> The momentum factor masking does not reset the momentum correction. It only blocks the momentum of delayed gradients from misleading the optimization.
>
> Suppose the last update is at iteration t-1, the next update at iteration t+T-1, and we only consider the gradients  { g_{t}, g_{t+1}, ..., g_{t+T-1} }
>
>          - Dense Update
>             w_{t+T} = w_{t} - lr x [ ... + (1 + m +  ... + m^{T-1}) x g_{t} +  (1 + m +  ... + m^{T-2}) x g_{t+1} + ... + 1 x g_{t+T-1}]
>             w_{t+\tau} = w_{t} - lr x [ ... + (1 + m +  ... + m^{\tau-1}) x g_{t} +  (1 + m +  ... + m^{\tau-2}) x g_{t+1} + ... + (1 + m +  ... + m^{\tau-T}) x g_{t+T-1} + ...], where \tau > T
>
>          - Only local gradient accumulation
>             the coefficients of  { g_{t}, g_{t+1}, ..., g_{t+T-1} } are always the same.
>             w_{t+T} = w_{t} - lr x [ ... + 1 x g_{t} +  1 x g_{t+1} + ... +  1 x g_{t+T-1}]
>             w_{t+\tau} = w_{t} - lr x [ ... + (1 + m + m^2 + ... + m^{\tau-T}) x (g_{t} + g_{t+1} + ... + g_{t+T-1}) + ...]
>
>          - With the momentum correction,
>             the coefficients of  { g_{t}, g_{t+1}, ..., g_{t+T-1} } are always the same as the dense update.
>             w_{t+T} = w_{t} - lr x [ ... + (1 + m +  ... + m^{T-1}) x g_{t} +  (1 + m +  ... + m^{T-2}) x g_{t+1} + ... + 1 x g_{t+T-1}]
>             w_{t+\tau} = w_{t} - lr x [ ... + (1 + m +  ... + m^{\tau-1}) x g_{t} +  (1 + m +  ... + m^{\tau-2}) x g_{t+1} + ... + (1 + m +  ... + m^{\tau-T}) x g_{t+T-1} + ...], where \tau > T
>
>          - With the momentum correction and momentum factor masking
>             we clear the local u_{t} to prevent the delayed gradients from misleading the optimization after they are used for the update.
>             w_{t+T} = w_{t} - lr x [ ... + (1 + m +  ... + m^{T-1}) x g_{t} +  (1 + m +  ... + m^{T-2}) x g_{t+1} + ... + 1 x g_{t+T-1}]
>             w_{t+\tau} = w_{t} - lr x [ ... + (1 + m +  ... + m^{T-1}) x g_{t} +  (1 + m +  ... + m^{T-2}) x g_{t+1} + ... + 1 x g_{t+T-1} + ...], where \tau > T

---

### Public Comment · (anonymous) · 2018-01-11
**Some details on CIFAR10 experiments**

Do the four rows in Table 2 (#GPUs in total = 4, 8, 16, 32) correspond to 1, 2, 4 and 8 training nodes? Could you please also say what is the compression ratio for these four cases? Thank you.

---

### Public Comment · ~Felix_Yu1 · 2018-01-29
**An earlier work on reducing communication cost in distributed SGD**

Dear authors,

I am writing to bring your attention to the following work
Suresh, Yu, Kumar, McMahan Distributed mean estimation with limited communication ICML 2017

In this paper, we showed a few variants to reduce the communication cost of distributed mean estimation. One method is communication optimal. The methods and analysis are directly applicable to distributed SGD, as each step of SGD involves distributed mean estimation of gradients.

Link to our theory paper (distributed mean estimation): (http://proceedings.mlr.press/v70/suresh17a/suresh17a.pdf) and link to our implementation workshop paper which you cited (https://arxiv.org/abs/1610.05492).

---

### Public Comment · (anonymous) · 2018-04-25
**Hi DeepGradientCompressionAuthors,**

Could you please provide more details of the derivation process from equation (3) to equation (4)  and from equation(5) to (6), I am having a hard rederivation on it to understand deeply, I got a different result, see [1] thanks.
[1]. https://www.mathcha.io/editor/n9j1S7QSeWtPBI68

---

> ### Author Response · Authors · 2018-04-30
> **Re: Hi DeepGradientCompressionAuthors**
>
> Thank you for your comments.
>
> First, in these two equations ((4) and (5)), we only explicitly exhibit the gradient $\triangledown_{k,t}$ calculated on the training node $k$, since they are what we care about.
>
> - from equation (3) to equation (4):
>   The first equation on [1] is the same as the equation (4), if you only look at the gradient $\triangledown_{k,t}$ calculated on the training node $k$.
>
> - from (5) to (6):
>   During the sparse update interval $T$, the gradient $\triangledown_{k,t}^{(i)}$ on the i-th position is never been sent, so $sparse(v_{k,t+T}^{(i)}) = v_{k,t+T}^{(i)} =  v_{k,t+T -1}^{(i)} + \triangledown_{k,t+T-1} = \cdots + \triangledown_{k,t+1} + \triangledown_{k,t}$ and $sparse(v_{k,t+T-1}^{(i)}) = sparse(v_{k,t+T-2}^{(i)}) = \cdots = 0 $.

---

### Public Comment · ~Ashiq_Imran1 · 2018-06-22
**Threshold value selection**

I was confused about selecting threshold values. Suppose, your gradients smallest values is 0.5 and highest values is 50.5, what should be s% of the gradient to select the threshold value.

---

### Public Comment · ~Chin_Woo1 · 2020-06-27
**nice**

Thank you and waiting for your new post
https://wordcounter.tools

---

### Decision · Program_Chairs · 2018-01-29
**ICLR 2018 Conference Acceptance Decision**

**Decision:**

Accept (Poster)

**Comment:**

This work proposes a hybrid system for large-scale distributed and federated training of commonly used deep networks. This problem is of broad interest and these methods have the potential to be significantly impactful, as is attested by the active and interesting discussion on this work. At first there were questions about the originality of this study, but it seems that the authors have now added extra references and comparisons.

Reviewers were split about the clarity of the paper itself. One notes that "on the whole clearly presented", but another finds it too dense, disorganized and needing of more clear explanation. Reviewers were also concerned that methods were a bit heuristic and could benefit from more details. There were also many questions about these details in the discussion forum, these should make it into the next version.  The main stellar aspect of the work were the experimental results, and reviewers call them "thorough" and note they are convincing.